# Decentralized Stochastic Control with Finite-Dimensional Memories: A Memory Limitation Approach

**DOI:** 10.3390/e25050791

**Published:** 2023-05-12

**Authors:** Takehiro Tottori, Tetsuya J. Kobayashi

**Affiliations:** 1Department of Mathematical Informatics, Graduate School of Information Science and Technology, The University of Tokyo, Tokyo 113-8654, Japan; 2Institute of Industrial Science, The University of Tokyo, Tokyo 153-8505, Japan; 3Department of Electrical Engineering and Information Systems, Graduate School of Engineering, The University of Tokyo, Tokyo 113-8654, Japan; 4Universal Biology Institute, The University of Tokyo, Tokyo 113-8654, Japan

**Keywords:** decision-making, optimal control, stochastic control, multi-agent system, decentralized control, memory limitation, mean-field control

## Abstract

Decentralized stochastic control (DSC) is a stochastic optimal control problem consisting of multiple controllers. DSC assumes that each controller is unable to accurately observe the target system and the other controllers. This setup results in two difficulties in DSC; one is that each controller has to memorize the infinite-dimensional observation history, which is not practical, because the memory of the actual controllers is limited. The other is that the reduction of infinite-dimensional sequential Bayesian estimation to finite-dimensional Kalman filter is impossible in general DSC, even for linear-quadratic-Gaussian (LQG) problems. In order to address these issues, we propose an alternative theoretical framework to DSC—memory-limited DSC (ML-DSC). ML-DSC explicitly formulates the finite-dimensional memories of the controllers. Each controller is jointly optimized to compress the infinite-dimensional observation history into the prescribed finite-dimensional memory and to determine the control based on it. Therefore, ML-DSC can be a practical formulation for actual memory-limited controllers. We demonstrate how ML-DSC works in the LQG problem. The conventional DSC cannot be solved except in the special LQG problems where the information the controllers have is independent or partially nested. We show that ML-DSC can be solved in more general LQG problems where the interaction among the controllers is not restricted.

## 1. Introduction

Optimal control problems of a stochastic dynamical system by decentralized multiple controllers appear in various practical applications, including real-time communication [1,2], decentralized hypothesis testing [3], and networked control [4]. Such problems have been extensively studied in stochastic optimal control theory as decentralized stochastic control (DSC) [5,6,7,8,9,10]. DSC consists of a target system and multiple controllers (Figure 1a) and assumes that each controller cannot completely observe the state of the system and the controls of the other controllers. The information of the target system and the other controllers was only obtained via their noisy observations. Thus, each controller should be optimized to determine its control solely from its own observation history. Even for a pair of finite-dimensional state and observation, the observation history is infinite-dimensional. As a result, ideally, the optimal controller obtained theoretically should possess infinite-dimensional memory. In practical applications, however, the available memory size of each controller is finite-dimensional and often severely limited. Thus, we have to obtain the solutions based on finite-dimensional memory by employing approximation methods heuristically, which may impair the optimality of the ideal solution, especially when the available memory size is not sufficient.

Moreover, another difficulty arises in DSC due to the decentralized setting. If the number of controllers is one, or if all controllers share their observation histories, DSC is reduced to partially observable stochastic control (POSC), in which the observation histories of all controllers can be summarized optimally as the posterior probability of the state by sequential Bayesian estimation [11,12,13,14,15,16]. The posterior probability of the state is also infinite-dimensional, and thus the same problem as DSC still survives, even for POSC. Nevertheless, in POSC, this difficulty can be circumvented by focusing only on the linear-quadratic-Gaussian (LQG) setting under which the posterior probability of the state can be represented by the finite-dimensional mean vector and covariance matrix of Gaussian distribution and the sequential Bayesian estimation can be computed by the Kalman filter [11,12,14]. Therefore, POSC is practically solved, at least in the LQG problem. The difficulty in DCS is that this nice property of LQG is not retained.

In DCS, each controller cannot access the observation histories of the other controllers, as well as the state of the system, which causes each controller to estimate all the others from their own observation history. This hampers the Bayesian estimation of the posterior to be computed sequentially and prevents the infinite-dimensional observation history from being compressed into the finite-dimensional sufficient statistics, even for the LQG problem. Some theoretical studies have addressed this issue by restricting the interaction among the controllers. If the information the controllers have is independent [8,9,10] or partially nested [17,18,19,20,21,22], DSC can enjoy the nice property of the LQG problem and be solved explicitly and optimally with finite-dimensional memory. However, the LQG problem with more general interactions as well as non-LQG problems are still an open problem in DSC.

In order to address these issues, we propose an alternative theoretical framework to DSC, memory-limited DSC (ML-DSC), which is the decentralized version of memory-limited POSC (ML-POSC) [23,24]. The two major difficulties in DSC originate from the ignorance of constraints over controllers when we derive the optimal estimation and control solution. Unlike conventional DSC, ML-DSC explicitly formulates finite-dimensional memories of the controllers and their capacities (Figure 1b). In ML-DSC, each controller is optimized to compress the infinite-dimensional observation history into the prescribed finite-dimensional memory and to determine the control based on it. In other words, each controller controls not only the dynamics of the target system but also the dynamics of its own memory. The formulation of ML-DSC enables us to evade the difficulties mentioned above.

Furthermore, we provide a way to solve the optimization problem associated with the ML-DSC formulation. Specifically, we address the optimization problem by converting ML-DSC in the state space into the deterministic optimal control problem in the probability density function space. This technique has recently been used in mean-field stochastic control [25,26] and ML-POSC [23,24], and is also effective for ML-DSC. Following that, we can solve ML-DSC in a similar way to the deterministic optimal control problem on the probability density function space; the optimal control function of ML-DSC was obtained by jointly solving the Hamilton–Jacobi–Bellman (HJB) equation and the Fokker–Planck (FP) equation. HJB–FP equations also appear in mean-field stochastic game and control [25,26,27,28,29] and ML-POSC [23,24], and numerous numerical algorithms have been proposed [24,30,31,32]. Using these numerical algorithms, ML-DSC may be solved effectively, even in general problems. It should be noted that a similar idea to ML-DSC was also employed in a decentralized partially observable Markov decision process (DEC-POMDP) with the finite-state controller for more than a decade [33,34,35,36,37,38,39]. However, the algorithms of the finite-state controller of DEC-POMDP strongly depend on the discreteness, and thus they are not applicable to ML-DSC where the continuous time and state are considered.

We applied ML-DSC and our algorithm to the LQG problem. The conventional DSC can only be solved for special LQG problems where the information of the controllers is independent [8,9,10] or partially nested [17,18,19,20,21,22]. In contrast, ML-DSC can be solved even in LQG problems with more general interactions among the controllers. In LQG problems of POSC, estimation and control are clearly separated, and are optimized by the Kalman filter and the Riccati equation, respectively [11,14]. In the LQG problems of ML-DSC, estimation and control are not clearly separated and are jointly optimized by the modified Riccati equation, which is called the decentralized Riccati equation in this paper. We noted that this coupling of estimation and control also appears in the conventional DSC [17,18,19,20,21,22] and ML-POSC [23,24]. Therefore, it may be induced by decentralized structure and memory limitation. Finally, we conducted two numerical experiments for the LQG problems of ML-DSC. One controls one-dimensional divergent state dynamics, and the other controls two-dimensional oscillatory state dynamics. These numerical experiments demonstrate that the decentralized Riccati equation is superior to the Riccati equation in the LQG problems of ML-DSC.

The rest of this paper is organized as follows. In Section 2, we briefly review the conventional DSC. In Section 3, we formulate ML-DSC. In Section 4, we solve ML-DSC. In Section 5, we apply ML-DSC to the LQG problem. In Section 6, we conduct the numerical experiments of two LQG problems in ML-DSC. In Section 7, we discuss this paper.

## 2. Review of Decentralized Stochastic Control

In this section, we briefly review the conventional DSC (Figure 1a) [8,9,10]. DSC consists of a target system and *N* controllers. Vector $x_{t}\in R^{d_{x}}$ is the state of the system at time t∈[0,T], which evolves by the following stochastic differential equation (SDE):(1)$$\begin{array}{cc} dx_{t} & =b\left(t,x_{t},u_{t}\right)dt+\sigma \left(t,x_{t},u_{t}\right)d\omega_{t}, \end{array}$$
where $x_{0}$ obeys $p_{0}\left(x_{0}\right)$, $\omega_{t}\in R^{d_{\omega }}$ is the standard Wiener process, $u_{t}^{i}\in R^{d_{u}^{i}}$ is the control of controller *i*, and $u_{t}:=\left(u_{t}^{1},u_{t}^{2},\ldots ,u_{t}^{N}\right)\in R^{d_{u}}$ is the joint control of all controllers. We noted that $d_{u}:=\sum_{i=1}^{N}d_{u}^{i}$. DSC often assumes that the system is composed of *N* agents and that the state of the system is decomposed into $x_{t}:=\left(x_{t}^{1},x_{t}^{2},\ldots ,x_{t}^{N}\right)\in R^{d_{x}}$ where $x_{t}^{i}\in R^{d_{x}^{i}}$ is the state of agent *i*. In this paper, we do not assume such a situation, because our formulation of the state of the system includes it as a special case.

In DSC, controller *i* cannot completely observe the state of the system $x_{t}$ and the joint control of all controllers $u_{t}$. It can only obtain the noisy observation $y_{t}^{i}\in R^{d_{y}^{i}}$, which evolves by the following SDE:(2)$$\begin{array}{cc} dy_{t}^{i} & =h^{i}\left(t,x_{t},u_{t}\right)dt+\gamma^{i}\left(t,x_{t},u_{t}\right)d\nu_{t}^{i}, \end{array}$$
where $y_{0}^{i}$ obeys $p_{0}^{i}\left(y_{0}^{i}\right)$, and $\nu_{t}^{i}\in R^{d_{\nu }^{i}}$ is the standard Wiener process. Controller *i*’s observation $y_{t}^{i}$ is controlled by the other controllers through the joint control $u_{t}$, which expresses the communication among the controllers. Controller *i* determines its control $u_{t}^{i}$ based on the observation history $y_{0:t}^{i}:=\left\{y_{\tau }^{i}|\tau \in \left[0,t\right]\right\}$ as follows:(3)$$\begin{array}{c} u_{t}^{i}=u^{i}\left(t,y_{0:t}^{i}\right). \end{array}$$

The objective function of DSC is given by the following expected cumulative cost function:(4)$$\begin{array}{c} J\left[u\right]:=E_{p(x_{0:T},y_{0:T};u)}\left[\int_{0}^{T}f\left(t,x_{t},u_{t}\right)dt+g\left(x_{T}\right)\right], \end{array}$$
where *f* is the running cost function, *g* is the terminal cost function, $p(x_{0:T},y_{0:T};u)$ is the joint probability of $x_{0:T}$ and $y_{0:T}$ given that *u* is a parameter, and $E_{p}\left[\cdot \right]$ is the expectation with respect to *p*. DSC is the problem to find the optimal joint control function $u^{*}$ that minimizes the objective function J[u]:(5)$$\begin{array}{c} u^{*}:=\arg\;\min_{u}J\left[u\right]. \end{array}$$

In DSC, controller *i* needs to memorize the infinite-dimensional observation history $y_{0:t}^{i}$ to determine the optimal control $u_{t}^{i*}=u^{i*}\left(t,y_{0:t}^{i}\right)$. This is one of the major obstacles in DSC for implementing controllers with finite and limited memory.

## 3. Memory-Limited Decentralized Stochastic Control

In this section, we formulate ML-DSC, which can circumvent the difficulty in DSC by explicitly formulating finite-dimensional memory of the controllers.

### 3.1. Problem Formulation

In this subsection, we formulate ML-DSC (Figure 1b). ML-DSC explicitly formulates the finite-dimensional memory of controller *i* by $z_{t}^{i}\in R^{d_{z}^{i}}$. The memory dimension $d_{z}^{i}$ is prescribed by the available memory size of the controller *i*. The controller *i* compresses the infinite-dimensional observation history $y_{0:t}^{i}$ into the finite-dimensional memory $z_{t}^{i}$ by the following SDE:(6)$$\begin{array}{c} dz_{t}^{i}=c^{i}\left(t,z_{t}^{i},v_{t}^{i}\right)dt+\kappa^{i}\left(t,z_{t}^{i},v_{t}^{i}\right)dy_{t}^{i}+\eta^{i}\left(t,z_{t}^{i},v_{t}^{i}\right)d\xi_{t}^{i}, \end{array}$$
where $z_{0}^{i}$ obeys $p_{0}^{i}\left(z_{0}^{i}\right)$, $\xi_{t}^{i}\in R^{d_{\xi }^{i}}$ is the standard Wiener process, and $v_{t}^{i}\in R^{d_{v}^{i}}$ is the control of the memory. Unlike the conventional DSC, ML-DSC can take into account the intrinsic stochasticity of the memory, which is modeled by the standard Wiener process $d\xi_{t}^{i}$ in the memory dynamics (Equation 6). In addition, the compression of the infinite-dimensional observation history $y_{0:t}^{i}$ into the finite-dimensional memory $z_{t}^{i}$ is optimized by the memory control $v_{t}^{i}$. In ML-DSC, the controller *i* determines the state control $u_{t}^{i}$ and the memory control $v_{t}^{i}$ based on the finite-dimensional memory $z_{t}^{i}$ as follows:(7)$$\begin{array}{c} u_{t}^{i}=u^{i}\left(t,z_{t}^{i}\right),\;v_{t}^{i}=v^{i}\left(t,z_{t}^{i}\right). \end{array}$$

The objective function of ML-DSC is given by the following expected cumulative cost function:(8)$$\begin{array}{c} J\left[u,v\right]:=E_{p(x_{0:T},y_{0:T},z_{0:T};u,v)}\left[\int_{0}^{T}f\left(t,x_{t},u_{t},v_{t}\right)dt+g\left(x_{T}\right)\right], \end{array}$$
where *f* is the running cost function, *g* is the terminal cost function, $p(x_{0:T},y_{0:T},z_{0:T};u,v)$ is the joint probability of $x_{0:T}$, $y_{0:T}$ and $z_{0:T}$ given *u* and *v* as parameters, and $E_{p}\left[\cdot \right]$ is the expectation with respect to *p*. Unlike the cost function *f* of DSC in Equation (Equation 4), the cost function *f* of ML-DSC in Equation (Equation 8) depends on the memory control $v_{t}$ as well as the state control $u_{t}$. From a practical viewpoint, it should be natural to consider both costs of control and memory. ML-DSC optimizes the state control function *u* and the memory control function *v* based on the objective function J[u,v]:(9)$$\begin{array}{c} u^{*},v^{*}:=\arg\;\min_{u,v}J\left[u,v\right]. \end{array}$$The optimal memory control function $v^{*}:=\left(v^{1*},v^{2*},\ldots ,v^{N*}\right)$ optimizes the memory dynamics (Equation 6), which can be interpreted as the optimization of the compression of the observation history into the finite-dimensional memory. In the LQG problem of POSC, the optimal memory control function $v^{*}$ makes the memory dynamics into the Kalman filter, which is the optimal compression of the observations history in this problem [23]. We expect that the optimal memory control function $v^{*}$ is also effective for more general problems of ML-DSC.

In ML-DSC, controller *i* determines the optimal control functions $u_{t}^{i*}$ and $v_{t}^{i*}$ based only on the finite-dimensional memory $z_{t}^{i}$. In addition, ML-DSC can take into account the intrinsic stochasticity and the control cost of the memory. Thus, ML-DSC can explicitly accommodate various realistic constrains of the controllers such as memory size, noise in the controllers, and cost for updating memory, none of which can be explicitly addressed in the conventional DSC.

It should be noted that here we consider memory size only for storing continuous time-series with finite-dimensional vectors. While memory size also matters when we consider quantization and storing of real valued observations, these topics are out of the scope of this work.

### 3.2. Extended State

In this subsection, we generalize the formulation of ML-DSC based on the extended state. This generalization is useful for mathematical investigations by simplifying the notation of ML-DSC. Furthermore, it clarifies the difference between ML-DSC and the conventional stochastic optimal control problems.

We define the extended state $s_{t}\in R^{d_{s}}$, the extended control ${\tilde{u}}_{t}^{i}\in R^{d_{\tilde{u}}^{i}}$, the extended joint control ${\tilde{u}}_{t}\in R^{d_{\tilde{u}}}$, and the extended standard Wiener process ${\tilde{\omega }}_{t}\in R^{d_{\tilde{\omega }}}$ as follows:(10)$$s_{t}:=\left(\begin{array}{c} x_{t}\\ z_{t}^{1}\\ \vdots \\ z_{t}^{N} \end{array}\right),\;\;{\tilde{u}}_{t}^{i}:=\left(\begin{array}{c} u_{t}^{i}\\ v_{t}^{i} \end{array}\right),\;\;{\tilde{u}}_{t}:=\left(\begin{array}{c} {\tilde{u}}_{t}^{1}\\ \vdots \\ {\tilde{u}}_{t}^{N} \end{array}\right),\;\;{\tilde{\omega }}_{t}:=\left(\begin{array}{c} \omega_{t}\\ \nu_{t}^{1}\\ \vdots \\ \nu_{t}^{N}\\ \xi_{t}^{1}\\ \vdots \\ \xi_{t}^{N} \end{array}\right),$$
where $d_{s}:=d_{x}+\sum_{i=1}^{N}d_{z}^{i}$, $d_{\tilde{u}}^{i}:=d_{u}^{i}+d_{v}^{i}$, $d_{\tilde{u}}:=\sum_{i=1}^{N}d_{\tilde{u}}^{i}$, and $d_{\tilde{\omega }}:=d_{\omega }+\sum_{i=1}^{N}d_{\nu }^{i}+\sum_{i=1}^{N}d_{\xi }^{i}$.

Based on the extended state $s_{t}$, the extended joint control ${\tilde{u}}_{t}$, and the extended standard Wiener process ${\tilde{\omega }}_{t}$, the state, observation, and memory SDEs, i.e., Equations (Equation 1), (Equation 2), and (Equation 6), are summarized as follows:(11)$$ds_{t}=\underset{=:\tilde{b}\left(t,s_{t},{\tilde{u}}_{t}\right)}{\underbrace{\left(\begin{array}{c} b\\ c^{1}+\kappa^{1}h^{1}\\ \vdots \\ c^{N}+\kappa^{N}h^{N} \end{array}\right)}}dt+\underset{=:\tilde{\sigma }\left(t,s_{t},{\tilde{u}}_{t}\right)}{\underbrace{\left(\begin{array}{ccccccc} \sigma & O & \ldots & O & O & \ldots & O\\ O & \kappa^{1}\gamma^{1} & \ldots & O & \eta^{1} & \ldots & O\\ \vdots & \vdots & \ddots & \vdots & \vdots & \ddots & \vdots \\ O & O & \ldots & \kappa^{N}\gamma^{N} & O & \ldots & \eta^{N} \end{array}\right)}}d{\tilde{\omega }}_{t},$$
where $p_{0}\left(s_{0}\right)=p_{0}\left(x_{0}\right)\prod_{i=1}^{N}p_{0}^{i}\left(z_{0}^{i}\right)$. Thus, the SDE of ML-DSC can be generalized as follows:(12)$$\begin{array}{c} ds_{t}=\tilde{b}\left(t,s_{t},{\tilde{u}}_{t}\right)dt+\tilde{\sigma }\left(t,s_{t},{\tilde{u}}_{t}\right)d{\tilde{\omega }}_{t}, \end{array}$$
where $s_{0}$ obeys $p_{0}\left(s_{0}\right)$. We note that the structures of $\tilde{b}\left(t,s_{t},{\tilde{u}}_{t}\right)$ and $\tilde{\sigma }\left(t,s_{t},{\tilde{u}}_{t}\right)$ in Equation (Equation 12) are not necessarily restricted to those in Equation (Equation 11). Importantly, in ML-DSC, controller *i* determines the extended control ${\tilde{u}}_{t}^{i}$ based solely on the memory $z_{t}^{i}$ as follows:(13)$$\begin{array}{c} {\tilde{u}}_{t}^{i}={\tilde{u}}^{i}\left(t,z_{t}^{i}\right). \end{array}$$

The objective function of ML-DSC (Equation 8) is generalized as follows:(14)$$\begin{array}{c} J\left[\tilde{u}\right]:=E_{p(s_{0:T};\tilde{u})}\left[\int_{0}^{T}\tilde{f}\left(t,s_{t},{\tilde{u}}_{t}\right)dt+\tilde{g}\left(s_{T}\right)\right], \end{array}$$
where $\tilde{f}$ is the running cost function and $\tilde{g}$ is the terminal cost function. Therefore, the generalized ML-DSC is the problem to find the optimal extended joint control function ${\tilde{u}}^{*}$ that minimizes the objective function $J[\tilde{u}]$:(15)$$\begin{array}{c} {\tilde{u}}^{*}:=\arg\;\min_{\tilde{u}}J\left[\tilde{u}\right] \end{array}$$
under the constraint of Equation (Equation 13).

This generalization (Equation 12)–(Equation 15) clarifies the difference between ML-DSC and the conventional stochastic optimal control problems. If controller *i* determines the extended control ${\tilde{u}}_{t}^{i}$ based on the whole extended state $s_{t}:=\left(x_{t},z_{t}^{1},\ldots ,z_{t}^{N}\right)$ as ${\tilde{u}}_{t}^{i}={\tilde{u}}^{i}\left(t,s_{t}\right)$, this problem becomes equivalent to completely observable stochastic control (COSC), which is the most basic stochastic optimal control problem (Figure 2a) [13,14,40]. Furthermore, if controller *i* determines the extended control ${\tilde{u}}_{t}^{i}$ based on the joint memory $z_{t}:=\left(z_{t}^{1},\ldots ,z_{t}^{N}\right)$ as ${\tilde{u}}_{t}^{i}={\tilde{u}}^{i}\left(t,z_{t}\right)$, this problem is reduced to ML-POSC in which all controllers share their information (Figure 2b) [23,24]. ML-DSC determines the extended control ${\tilde{u}}_{t}^{i}$ based solely on its own memory $z_{t}^{i}$ as ${\tilde{u}}_{t}^{i}={\tilde{u}}^{i}\left(t,z_{t}^{i}\right)$ (Equation 13), which is different from COSC and ML-POSC (Figure 2c). While ML-DSC cannot be solved in a similar way as COSC [14,40,41], as is shown in the next section, it can be solved in a similar way as ML-POSC [23,24] because the method of ML-POSC is more general than that of COSC.

In the following section, we mainly consider the formulation of this subsection rather than that of Section 3.1 because it is simpler and more general. Moreover, we omit $\tilde{\cdot }$ for the notational simplicity.

## 4. Derivation of Optimal Control Function

In this section, we solve ML-DSC by employing the technique in mean-field stochastic control [25,26] and ML-POSC [23,24].

### 4.1. Derivation of Optimal Control Function

In this subsection, we derive the optimal control function of ML-DSC. In ML-DSC, each controller cannot directly access the information about the state of the system and the memories of the other controllers. This constraint makes ML-DSC unable to be solved by the conventional methods of COSC, such as Bellman’s dynamic programming principle on the extended state space [13,14,40]. In order to address this issue, we converted ML-DSC on the extended state space into the deterministic optimal control problem on the probability density function space. The similar technique has also been used in mean-field stochastic control [25,26] and ML-POSC [23,24], and it is more effective for a broader class of stochastic optimal control problems than the conventional methods of COSC.

The extended state SDE (Equation 12) can be converted into the following Fokker–Planck (FP) equation:(16)$$\begin{array}{c} \frac{\partial p_{t}\left(s\right)}{\partial t}=L_{u}^{†}p_{t}\left(s\right), \end{array}$$
where the initial condition is given by $p_{0}\left(s\right)$, and $L_{u}^{†}$ is the forward diffusion operator, which is defined by
(17)$$\begin{array}{cc} L_{u}^{†}p_{t}\left(s\right) & :=-\sum_{i=1}^{d_{s}}\frac{\partial (b_{i}\left(t,s,u\right)p_{t}\left(s\right))}{\partial s_{i}}+\frac{1}{2}\sum_{i,j=1}^{d_{s}}\frac{\partial^{2}\left(D_{ij}\left(t,s,u\right)p_{t}\left(s\right)\right)}{\partial s_{i}\partial s_{j}}, \end{array}$$
where $D\left(t,s,u\right):=\sigma \left(t,s,u\right)\sigma^{⊤}\left(t,s,u\right)$. The objective function (Equation 14) can be calculated as follows:(18)$$\begin{array}{c} J\left[u\right]=\int_{0}^{T}\bar{f}\left(t,p_{t},u_{t}\right)dt+\bar{g}\left(p_{T}\right), \end{array}$$
where $\bar{f}\left(t,p,u\right):=E_{p(s)}\left[f\left(t,s,u\right)\right]$ and $\bar{g}\left(p\right):=E_{p(s)}\left[g\left(s\right)\right]$. We note that $\tilde{\cdot }$ is omitted for the notational simplicity. From Equations (Equation 16) and (Equation 18), ML-DSC on the extended state space is converted into the deterministic optimal control problem on the probability density function space.

If being represented by the extended state, each controller cannot completely access the information of the extended state in ML-DSC, which hampers the conventional methods of COSC. By lifting the state variable from the extended state to its probability density function, such a difficulty can be avoided, because any controllers can completely access the probability density function from its deterministic nature. As a result, the optimal condition of ML-DSC was obtained by a similar way to the deterministic optimal control problem, i.e., Pontryagin’s minimum principle on the probability density function, which can be interpreted as the generalization of Bellman’s dynamic programming principle on the extended state space [23,24]:

**Theorem** **1.**
*The optimal control function of ML-DSC satisfies the following equation:*

(19)
$$\begin{array}{c} u^{i*}\left(t,z^{i}\right)=\arg\;\min_{u^{i}}E_{p_{t}\left(s^{-i}|z^{i}\right)}\left[H\left(t,s,(u^{-i*},u^{i}),w\right)\right],{\;}^{\forall }i\in \left\{1,2,\ldots ,N\right\}, \end{array}$$

*where H is the Hamiltonian, which is defined as follows:*

(20)
$$\begin{array}{c} H\left(t,s,u,w\right):=f\left(t,s,u\right)+L_{u}w\left(t,s\right), \end{array}$$

*where $L_{u}$ is the backward diffusion operator, which is defined as follows:*

(21)
$$\begin{array}{cc} L_{u}w\left(t,s\right) & :=\sum_{i=1}^{d_{s}}b_{i}\left(t,s,u\right)\frac{\partial w(t,s)}{\partial s_{i}}+\frac{1}{2}\sum_{i,j=1}^{d_{s}}D_{ij}\left(t,s,u\right)\frac{\partial^{2}w\left(t,s\right)}{\partial s_{i}\partial s_{j}}, \end{array}$$

*which is the conjugate of $L_{u}^{†}$ as follows:*

(22)
$$\begin{array}{c} \int w\left(t,s\right)L_{u}^{†}p\left(t,s\right)ds=\int p\left(t,s\right)L_{u}w\left(t,s\right)ds. \end{array}$$

*Variables $s^{-i}\in R^{d_{s}^{-i}}$, $u^{-i*}\in R^{d_{u}^{-i}}$, and $\left(u^{-i*},u^{i}\right)\in R^{d_{u}}$ are defined as follows:*

(23)
$$s^{-i}:=\left(\begin{array}{c} x\\ z^{1}\\ \vdots \\ z^{i-1}\\ z^{i+1}\\ \vdots \\ z^{N} \end{array}\right),\;\;u^{-i*}:=\left(\begin{array}{c} u^{1*}\\ \vdots \\ u^{i-1*}\\ u^{i+1*}\\ \vdots \\ u^{N*} \end{array}\right),\;\;\left(u^{-i*},u^{i}\right):=\left(\begin{array}{c} u^{1*}\\ \vdots \\ u^{i-1*}\\ u^{i}\\ u^{i+1*}\\ \vdots \\ u^{N*} \end{array}\right),$$

*where $d_{s}^{-i}:=d_{s}-d_{z}^{i}$ and $d_{u}^{-i}:=d_{u}-d_{u}^{i}$. Function w(t,s) is the solution of the following Hamilton–Jacobi–Bellman (HJB) equation:*

(24)
$$\begin{array}{c} -\frac{\partial w(t,s)}{\partial t}=H\left(t,s,u^{*},w\right), \end{array}$$

*where w(T,s)=g(s). Function $p_{t}\left(s^{-i}|z^{i}\right):=p_{t}\left(s\right)/\int p_{t}\left(s\right)ds^{-i}$ is the conditional probability density function of $s^{-i}$ given $z^{i}$, and $p_{t}\left(s\right)$ is the solution of FP Equation (Equation 16) driven by $u^{*}$. We note that $\tilde{\cdot }$ is omitted for the notational simplicity.*


**Proof.** The proof is shown in Appendix A. □

We note that the optimality condition (Equation 19) is a necessary condition of the optimal control function of ML-DSC, not a sufficient condition. The optimality condition (Equation 19) becomes a necessary and sufficient condition when the expected Hamiltonian $\bar{H}\left(t,p,u,w\right):=E_{p(s)}\left[H\left(t,s,u,w\right)\right]$ is convex with respect to *p* and *u*. This proof is almost the same with Reference [24]. In the following, the control function of ML-DSC that satisfies the optimality condition (Equation 19) is called the optimal control function of ML-DSC.

### 4.2. Numerical Algorithm

The optimal control function of ML-DSC (Equation 19) is obtained by jointly solving FP Equation (Equation 16) and HJB Equation (Equation 24). HJB-FP equations also appear in mean-field stochastic game and control [25,26,27,28,29] and ML-POSC [23,24], and numerous numerical algorithms have been developed [24,32]. As a result, ML-DSC may be solved practically by employing these numerical algorithms.

One of the most basic numerical algorithms to solve HJB-FP equations is the forward-backward sweep method (the fixed-point iteration method) [24,32,42,43,44], which computes the forward FP Equation (Equation 16) and the backward HJB Equation (Equation 24) alternately. While the convergence of the forward-backward sweep method is not guaranteed in mean-field stochastic game and control [32,42,43,44], it is guaranteed in ML-POSC because the coupling of HJB-FP equations is limited to the optimal control function in ML-POSC [24]. The convergence of the forward-backward sweep method is also guaranteed in ML-DSC for the same reason as ML-POSC. This proof is almost the same as Reference [24].

### 4.3. Comparison with Completely Observable Stochastic Control or Memory-Limited Partially Observable Stochastic Control

COSC and ML-POSC can be solved in a similar way to ML-DSC [23,24]. In COSC, controller *i* can completely observe the state of the target system $x_{t}$ and the memories of the other controllers $z_{t}^{j}\;\left(j\neq i\right)$, as well as its own memory $z_{t}^{i}$ (Figure 2a) [13,14,40]. As a result, the control $u_{t}^{i}$ is determined based on the whole extended state $s_{t}:=\left(x_{t},z_{t}^{1},\ldots ,z_{t}^{N}\right)$ as $u_{t}^{i}=u^{i}\left(t,s_{t}\right)$. From Pontryagin’s minimum principle on the probability density function space, the optimal control function of COSC is given by the following equation:(25)$$\begin{array}{c} u^{i*}\left(t,s\right)=\arg\;\min_{u^{i}}H\left(t,s,(u^{-i*},u^{i}),w\right), \end{array}$$
where w(t,s) is the solution of HJB Equation (Equation 24). This result is the same with Bellman’s dynamic programming principle on the state space. Thus, Pontryagin’s minimum principle on the probability density function space can be interpreted as the generalization of Bellman’s dynamic programming principle on the state space.

In ML-POSC, controller *i* can observe the memories of the other controllers $z_{t}^{j}\;\left(j\neq i\right)$ as well as its own memory $z_{t}^{i}$ (Figure 2b) [23,24]. As a result, the control $u_{t}^{i}$ is determined based on the joint memory $z_{t}:=\left(z_{t}^{1},\ldots ,z_{t}^{N}\right)$ as $u_{t}^{i}=u^{i}\left(t,z_{t}\right)$. From Pontryagin’s minimum principle on the probability density function space, the optimal control function of ML-POSC is given by the following equation:(26)$$\begin{array}{c} u^{i*}\left(t,z\right)=\arg\;\min_{u^{i}}E_{p_{t}\left(x|z\right)}\left[H\left(t,s,(u^{-i*},u^{i}),w\right)\right], \end{array}$$
where w(t,s) is the solution of HJB Equation (Equation 24). $p_{t}\left(x|z\right):=p_{t}\left(s\right)/\int p_{t}\left(s\right)dx$ is the conditional probability density function of *x* given *z*, and $p_{t}\left(s\right)$ is the solution of FP Equation (Equation 16) driven by $u^{*}$.

Although HJB Equation (Equation 24) is the same for COSC, ML-POSC, and ML-DSC, the optimal control function is different. Notably, the optimal control functions of ML-POSC and ML-DSC depend on FP Equation (Equation 16) because they need to estimate unobservables from observables. ML-POSC needs to estimate the state of the system $x_{t}$ from the joint memory of all controllers $z_{t}$. ML-DSC needs to estimate the memories of the other controllers $z_{t}^{j}\;\left(j\neq i\right)$ as well as the state of the system $x_{t}$ from its own memory $z_{t}^{i}$.

In COSC, the optimal control function depends only on HJB Equation (Equation 24). As a result, it can be obtained by solving the HJB Equation (Equation 24) backward in time from the terminal condition, which is called the value iteration method [45,46,47]. By contrast, in ML-POSC and ML-DSC, the optimal control function cannot be obtained by the value iteration method because it depends on FP Equation (Equation 16) as well as HJB Equation (Equation 24). Instead, it can be obtained by the forward-backward sweep method, which computes the forward FP Equation (Equation 16) and the backward HJB Equation (Equation 24) alternately [24].

## 5. Linear-Quadratic-Gaussian Problem

In this section, we demonstrate how ML-DSC works by applying it to the LQG problem. In the conventional DSC, the LQG problem can be solved only when the information of the controllers is independent [8,9,10] or partially nested [17,18,19,20,21,22]. By contrast, in ML-DSC, the LQG problem can be solved without restricting the interaction among the controllers.

### 5.1. Problem Formulation

In this subsection, we formulate the LQG problem of ML-DSC. In this problem, the extended state SDE (Equation 12) is given as follows:(27)$$\begin{array}{cc} ds_{t} & =\left(A\left(t\right)s_{t}+B\left(t\right)u_{t}\right)dt+\sigma \left(t\right)d\omega_{t}\\ \; & =\left(A\left(t\right)s_{t}+\sum_{i=1}^{N}B_{i}\left(t\right)u_{t}^{i}\right)dt+\sigma \left(t\right)d\omega_{t}, \end{array}$$
where the initial condition is given by the Gaussian distribution $p_{0}\left(s\right):=N\left(s\left|\mu_{0},\sum_{0}\right.\right)$. Furthermore, we note that $\tilde{\cdot }$ is omitted for the notational simplicity. In ML-DSC, controller *i* determines the control $u_{t}^{i}$ based on the memory $z_{t}^{i}$ as follows:(28)$$\begin{array}{c} u_{t}^{i}=u^{i}\left(t,z_{t}^{i}\right). \end{array}$$

In the extended state SDE (Equation 27), the interaction among the controllers is not restricted. For example, each controller is allowed to control the memories of the other controllers from its own memory through the state or the observations. In this case, the memories of the controllers do not become independent or partially nested. It becomes obvious in the numerical experiments in Section 6.

The objective function (Equation 14) is given as follows:(29)$$\begin{array}{cc} J[u] & :=E_{p(s_{0:T};u)}\left[\int_{0}^{T}\left(s_{t}^{⊤}Q\left(t\right)s_{t}+u_{t}^{⊤}R\left(t\right)u_{t}\right)dt+s_{T}^{⊤}Ps_{T}\right]\\ \; & =E_{p(s_{0:T};u)}\left[\int_{0}^{T}\left(s_{t}^{⊤}Q\left(t\right)s_{t}+\sum_{i=1}^{N}\sum_{j=1}^{N}{\left(u_{t}^{i}\right)}^{⊤}R_{ij}\left(t\right)u_{t}^{j}\right)dt+s_{T}^{⊤}Ps_{T}\right], \end{array}$$
where Q(t)⪰O, R(t)≻O, and P⪰O. The objective of this problem is to find the optimal control function $u^{*}$ that minimizes the objective function J[u]:(30)$$\begin{array}{c} u^{*}:=\arg\;\min_{u}J\left[u\right]. \end{array}$$

In this paper, we assume that R(t) is the block diagonal matrix, i.e., the costs of different controllers are independent as follows:(31)$$\begin{array}{c} R\left(t\right)=\left(\begin{array}{cccc} R_{11}\left(t\right) & O & \ldots & O\\ O & R_{22}\left(t\right) & \ldots & O\\ \vdots & \vdots & \ddots & \vdots \\ O & O & \ldots & R_{NN}\left(t\right) \end{array}\right), \end{array}$$
where $R_{ii}\left(t\right)≻O$. In this case, the objective function (Equation 29) can be calculated as follows:(32)$$\begin{array}{cc} J[u] & =E_{p(s_{0:T};u)}\left[\int_{0}^{T}\left(s_{t}^{⊤}Q\left(t\right)s_{t}+\sum_{i=1}^{N}{\left(u_{t}^{i}\right)}^{⊤}R_{ii}\left(t\right)u_{t}^{i}\right)dt+s_{T}^{⊤}Ps_{T}\right]. \end{array}$$If this assumption does not hold, the optimal control function cannot be derived explicitly. This problem is similar with the Witsenhausen’s counterexample, which demonstrates the difficulty of DSC and is historically recognized as an important problem [5,48,49]. However, this assumption is not critical in many applications as the control cost matrix is usually diagonal.

### 5.2. Derivation of Optimal Control Function

In this subsection, we derive the optimal control function of the LQG problem of ML-DSC by applying Theorem 1. In this problem, the probability density function of the extended state *s* at time *t* is given by the Gaussian distribution $p_{t}\left(s\right):=N\left(s|\mu (t),\sum (t)\right)$, in which μ(t) is the mean vector and ∑(t) is the covariance matrix. Defining the stochastic extended state $\hat{s}:=s-\mu$, the conditional expectation $E_{p_{t}\left(s^{-i}|z^{i}\right)}\left[s\right]$ can be calculated as follows:(33)$$\begin{array}{c} E_{p_{t}\left(s^{-i}|z^{i}\right)}\left[s\right]=\mu \left(t\right)+K_{i}\left(t\right)\hat{s}, \end{array}$$
where $K_{i}\left(t\right)\in R^{d_{s}\times d_{s}}$ is defined as follows:(34)$$\begin{array}{c} K_{i}\left(t\right):=\left(\begin{array}{ccccccc} O & \ldots & O & \sum_{xz^{i}}\left(t\right)\sum_{z^{i}z^{i}}^{-1}\left(t\right) & O & \ldots & O\\ O & \ldots & O & \sum_{z^{1}z^{i}}\left(t\right)\sum_{z^{i}z^{i}}^{-1}\left(t\right) & O & \ldots & O\\ \vdots & \ddots & \vdots & \vdots & \vdots & \ddots & \vdots \\ O & \ldots & O & \sum_{z^{i-1}z^{i}}\left(t\right)\sum_{z^{i}z^{i}}^{-1}\left(t\right) & O & \ldots & O\\ O & \ldots & O & I & O & \ldots & O\\ O & \ldots & O & \sum_{z^{i+1}z^{i}}\left(t\right)\sum_{z^{i}z^{i}}^{-1}\left(t\right) & O & \ldots & O\\ \vdots & \ddots & \vdots & \vdots & \vdots & \ddots & \vdots \\ O & \ldots & O & \sum_{z^{N}z^{i}}\left(t\right)\sum_{z^{i}z^{i}}^{-1}\left(t\right) & O & \ldots & O \end{array}\right). \end{array}$$Function $K_{i}\left(t\right)$ is the zero matrix except for the columns corresponding to $z^{i}$. By applying Theorem 1 to the LQG problem of ML-DSC, we obtained the following theorem:

**Theorem** **2.**
*In the LQG problem, the optimal control function of ML-DSC satisfies the following equation:*

(35)
$$\begin{array}{c} u^{i*}\left(t,z^{i}\right)=-R_{ii}^{-1}B_{i}^{⊤}\left(\Psi \mu +\Phi K_{i}\hat{s}\right), \end{array}$$

*where $K_{i}\left(t\right)$ is defined by Equation (Equation 34), which depends on ∑(t). Functions μ(t) and ∑(t) are the solutions of the following ordinary differential equations (ODEs):*

(36)
$$\begin{array}{cc} \frac{d\mu }{dt} & =\left(A-BR^{-1}B^{⊤}\Psi \right)\mu , \end{array}$$


(37)
$$\begin{array}{cc} \frac{d\sum }{dt} & =\sigma \sigma^{⊤}+\left(A-\sum_{i=1}^{N}B_{i}R_{ii}^{-1}B_{i}^{⊤}\Phi K_{i}\right)\sum +\sum {\left(A-\sum_{i=1}^{N}B_{i}R_{ii}^{-1}B_{i}^{⊤}\Phi K_{i}\right)}^{⊤}, \end{array}$$

*where $\mu \left(0\right)=\mu_{0}$ and $\sum \left(0\right)=\sum_{0}$. Functions Ψ(t) and Φ(t) are the solutions of the following ODEs:*

(38)
$$\begin{array}{cc} -\frac{d\Psi }{dt} & =Q+A^{⊤}\Psi +\Psi A-\Psi BR^{-1}B^{⊤}\Psi , \end{array}$$


(39)
$$\begin{array}{cc} -\frac{d\Phi }{dt} & =Q+A^{⊤}\Phi +\Phi A-\Phi BR^{-1}B^{⊤}\Phi +\sum_{i=1}^{N}{\left(I-K_{i}\right)}^{⊤}\Phi B_{i}R_{ii}^{-1}B_{i}^{⊤}\Phi \left(I-K_{i}\right), \end{array}$$

*where Ψ(T)=Φ(T)=P.*


**Proof.** The proof is shown in Appendix B. □

In the LQG problem of ML-DSC, FP Equation (Equation 16) is reduced to Equations (Equation 36) and (Equation 37), and HJB Equation (Equation 24) is reduced to Equations (Equation 38) and (Equation 39). The optimal control function (Equation 35) is decomposed into the deterministic part and the stochastic part, which correspond to the first term and the second term, respectively. The first term of the optimal control function (Equation 35) also appears in the linear-quadratic (LQ) problem of deterministic control, and Equation (Equation 38) is called the Riccati equation [14,40]. In contrast, the second term of the optimal control function (Equation 35) is new in the LQG problem of ML-DSC, and Equation (Equation 39) is called the decentralized Riccati equation in this paper.

### 5.3. Comparison with Completely Observable Stochastic Control or Memory-Limited Partially Observable Stochastic Control

In the LQG problem, the optimal control function of COSC is given as follows [14,40]:(40)$$\begin{array}{c} u^{i*}\left(t,s\right)=-R_{ii}^{-1}B_{i}^{⊤}\left(\Psi \mu +\Psi \hat{s}\right), \end{array}$$
where Ψ(t) is the solution of the Riccati Equation (Equation 38). Function Ψ(t) appears in the second term and the first term in COSC. In addition, the optimal control function of ML-POSC is given as follows [23,24]:(41)$$\begin{array}{c} u^{i*}\left(t,z\right)=-R_{ii}^{-1}B_{i}^{⊤}\left(\Psi \mu +\Pi K\hat{s}\right), \end{array}$$
where Π(t) is the solution of the following partially observable Riccati equation:(42)$$\begin{array}{cc} -\frac{d\Pi }{dt} & =Q+A^{⊤}\Pi +\Pi A-\Pi BR^{-1}B^{⊤}\Pi +{\left(I-K\right)}^{⊤}\Pi BR^{-1}B^{⊤}\Pi \left(I-K\right), \end{array}$$
where Π(T)=P and K(t) is defined by
(43)$$\begin{array}{c} K\left(t\right):=\left(\begin{array}{cc} O & \sum_{xz}\left(t\right)\sum_{zz}^{-1}\left(t\right)\\ O & I \end{array}\right). \end{array}$$Thus, the decentralized Riccati Equation (Equation 39) is a natural extension of the partially observable Riccati Equation (Equation 42) from a centralized problem to a decentralized problem.

While the first deterministic term of the optimal control function is the same for COSC (Equation 40), ML-POSC (Equation 41), and ML-DSC (Equation 35), the second stochastic term is different, reflecting the fact that the observable part of the stochastic extended state is different.

### 5.4. Decentralized Riccati Equation

In this subsection, we analyze the decentralized Riccati Equation (Equation 39) by comparing it with the Riccati Equation (Equation 38). The Riccati Equation (Equation 38) is the control gain matrix of deterministic control and COSC. While the Riccati Equation (Equation 38) optimizes control, it does not improve estimation, because estimation is not needed in deterministic control and COSC. In contrast, the decentralized Riccati Equation (Equation 39) may improve estimation as well as control because the controllers of ML-DSC need to estimate the state of the system and the memories of the other controllers from their own memories.

In order to support this discussion, we analyze the last term of the decentralized Riccati Equation (Equation 39), which is denoted as follows:(44)$$\begin{array}{c} Q_{i}:={\left(I-K_{i}\right)}^{⊤}\Phi B_{i}R_{ii}^{-1}B_{i}^{⊤}\Phi \left(I-K_{i}\right). \end{array}$$This term is the main difference between the Riccati Equation (Equation 38) and the decentralized Riccati Equation (Equation 39), and thus accounts for the contribution of estimation in ML-DSC. For the sake of simplicity, we focused on $Q_{N}$. A similar discussion is possible for $Q_{i}$ by the permutation of the controllers’ indices. We also denoted $a:=s^{-N}$ and $b:=z^{N}$ for notational simplicity. *a* is unobservable and *b* is observable for the controller *N*. $Q_{N}$ can be calculated as follows:(45)$$\begin{array}{c} Q_{N}=\left(\begin{array}{cc} P_{aa} & -P_{aa}\sum_{ab}\sum_{bb}^{-1}\\ -\sum_{bb}^{-1}\sum_{ba}P_{aa} & \sum_{bb}^{-1}\sum_{ba}P_{aa}\sum_{ab}\sum_{bb}^{-1} \end{array}\right), \end{array}$$
where $P_{aa}:={\left(\Phi B_{N}R_{NN}^{-1}B_{N}^{⊤}\Phi \right)}_{aa}$. Because $P_{aa}⪰O$ and $\sum_{bb}^{-1}\sum_{ba}P_{aa}\sum_{ab}\sum_{bb}^{-1}⪰O$, $\Phi_{aa}$ and $\Phi_{bb}$ may be larger than $\Psi_{aa}$ and $\Psi_{bb}$, respectively. Because $\Phi_{aa}$ and $\Phi_{bb}$ are the negative feedback gains of *a* and *b*, respectively, $Q_{N}$ may contribute to decreasing $\sum_{aa}$ and $\sum_{bb}$. Moreover, when $\sum_{ab}$ is positive/negative, $\Phi_{ab}$ may be smaller/larger than $\Psi_{ab}$, which may increase/decrease $\sum_{ab}$. The similar discussion is possible for $\sum_{ba}$, $\Phi_{ba}$, and $\Psi_{ba}$ because ∑, Φ, and Ψ are symmetric matrices. As a result, $Q_{N}$ may contribute to decreasing the following conditional covariance matrix:(46)$$\begin{array}{c} \sum_{a|b}:=\sum_{aa}-\sum_{ab}\sum_{bb}^{-1}\sum_{ba}, \end{array}$$
which corresponds to the estimation error of the unobservable *a* from the observable *b*. It indicates that the decentralized Riccati Equation (Equation 39) may improve estimation as well as control.

It should be noted that estimation and control are not clearly separated in the LQG problem of ML-DSC. In the LQG problem of POSC, estimation and control are clearly separated, and they are optimized by the Kalman filter and the Riccati equation, respectively [11,14]. By contrast, in the LQG problem of ML-DSC, both estimation and control are optimized by the decentralized Riccati equation. This coupling of estimation and control also appears in the conventional DSC [17,18,19,20,21,22] and ML-POSC [23,24], which indicates that it may be caused by a decentralized structure and memory limitation.

## 6. Numerical Experiments

In this section, we demonstrate the significance of the decentralized Riccati Equation (Equation 39) by conducting the numerical experiments of two LQG problems in ML-DSC. One is the one-dimensional state case, and the other is the two-dimensional state case.

### 6.1. One-Dimensional State Case

In this subsection, we conduct a numerical experiment in one-dimensional state case (Figure 3a). In this case, we consider the state $x_{t}\in R$, the observation $y_{t}^{i}\in R$ and the memory $z_{t}^{i}\in R$ of the controller i∈{1,2}, which evolve by the following SDEs: (47)$$\begin{array}{cc} dx_{t} & =\left(x_{t}+u_{x,t}^{1}+u_{x,t}^{2}\right)dt+d\omega_{t}, \end{array}$$(48)$$\begin{array}{cc} dy_{t}^{1} & =\left(x_{t}+u_{y,t}^{2}\right)dt+d\nu_{t}^{1}, \end{array}$$(49)$$\begin{array}{cc} dy_{t}^{2} & =\left(x_{t}+u_{y,t}^{1}\right)dt+d\nu_{t}^{2}, \end{array}$$(50)$$\begin{array}{cc} dz_{t}^{1} & =v_{t}^{1}dt+dy_{t}^{1}, \end{array}$$(51)$$\begin{array}{cc} dz_{t}^{2} & =v_{t}^{2}dt+dy_{t}^{2}. \end{array}$$The initial conditions are given by the standard Gaussian distributions. $\omega_{t}\in R$, $\nu_{t}^{1}\in R$, and $\nu_{t}^{2}\in R$ are the independent standard Wiener processes. $u_{t}^{i}:=\left(u_{x,t}^{i},u_{y,t}^{i}\right)=u^{i}\left(t,z_{t}^{i}\right)\in R^{2}$ and $v_{t}^{i}:=v^{i}\left(t,z_{t}^{i}\right)\in R$ are the controls of controller *i*. Each controller can control the memory of the other controller through $u_{y,t}^{i}$, which can be interpreted as the communication among the controllers. The objective function to be minimized is given as follows:(52)$$\begin{array}{c} J\left[u,v\right]:=E_{p(x_{0:T},y_{0:T},z_{0:T};u,v)}\left[\int_{0}^{10}\left(x_{t}^{2}+\sum_{i=1}^{2}\left({\left(u_{x,t}^{i}\right)}^{2}+{\left(u_{y,t}^{i}\right)}^{2}+{\left(v_{t}^{i}\right)}^{2}\right)\right)dt\right], \end{array}$$
where $u:=(u^{1},u^{2})$ and $v:=(v^{1},v^{2})$. Therefore, the objective of this problem is to minimize the state variance by small controls (Figure 3b).

In this problem, the information of the controllers is neither independent nor partially nested. The information of the controller 1’s memory $z_{t}^{1}$ propagates to the controller 2’s memory $z_{t}^{2}$ through the state control $u_{x,t}^{1}$ and the observation control $u_{y,t}^{1}$, and vice versa (Figure 3a). While such a problem cannot be solved by the conventional DSC, ML-DSC can address it.

**Figure 3 entropy-25-00791-f003:**
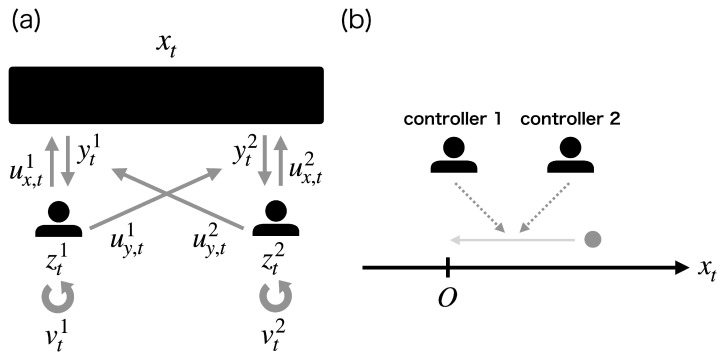
Schematic diagram of the LQG problem in Section 6.1. (**a**) The state of the system $x_{t}$ is one-dimensional. (**b**) The two controllers control the state of the system to be close to 0.

In the representation using the extended state $s_{t}:=\left(x_{t},z_{t}^{1},z_{t}^{2}\right)\in R^{3}$, the extended control ${\tilde{u}}_{t}^{i}:=\left(u_{x,t}^{i},u_{y,t}^{i},v_{t}^{i}\right)={\tilde{u}}^{i}\left(t,z_{t}^{i}\right)\in R^{3}$, and the extended standard Wiener process ${\tilde{\omega }}_{t}:=\left(\omega_{t},\nu_{t}^{1},\nu_{t}^{2}\right)\in R^{3}$ as in Equations (Equation 27) and (Equation 32), the SDEs defined by Equations (Equation 47)–(Equation 51) can be described as follows:(53)$$ds_{t}=\left(\left(\begin{array}{ccc} 1 & 0 & 0\\ 1 & 0 & 0\\ 1 & 0 & 0 \end{array}\right)s_{t}+\left(\begin{array}{ccc} 1 & 0 & 0\\ 0 & 0 & 1\\ 0 & 1 & 0 \end{array}\right){\tilde{u}}_{t}^{1}+\left(\begin{array}{ccc} 1 & 0 & 0\\ 0 & 1 & 0\\ 0 & 0 & 1 \end{array}\right){\tilde{u}}_{t}^{2}\right)dt+d{\tilde{\omega }}_{t},$$
which corresponds to Equation (Equation 27). The objective function (Equation 52) can be rewritten as follows:(54)$$J\left[\tilde{u}\right]:=E_{p(s_{0:T};\tilde{u})}\left[\int_{0}^{10}\left(s_{t}^{⊤}\left(\begin{array}{ccc} 1 & 0 & 0\\ 0 & 0 & 0\\ 0 & 0 & 0 \end{array}\right)s_{t}+\sum_{i=1}^{2}{\left({\tilde{u}}_{t}^{i}\right)}^{⊤}\left(\begin{array}{ccc} 1 & 0 & 0\\ 0 & 1 & 0\\ 0 & 0 & 1 \end{array}\right){\tilde{u}}_{t}^{i}\right)dt\right],$$
which corresponds to Equation (Equation 32). In addition, it satisfies the block diagonal matrix assumption of R(t) (Equation 31).

The Riccati Equation (Equation 38) can be solved backwards in time from the terminal condition. In contrast, the decentralized Riccati Equation (Equation 39) cannot be solved in a similar way to the Riccati Equation (Equation 38) because it depends on the covariance matrix ∑(t) via the estimation gain matrix $K_{i}\left(t\right)$, which is the solution of the time-forward ODE (Equation 37). In order to solve the decentralized Riccati Equation (Equation 39), which is the time-backward ODE of Φ(t), we use the forward-backward sweep method, which computes the time-forward ODE of ∑(t) (Equation 37) and the time-backward ODE of Φ(t) (Equation 39) alternately [24]. We note that the partially observable Riccati Equation (Equation 42) can also be solved by the forward-backward sweep method [24].

Figure 4 shows the trajectories of Ψ(t), Π(t), and Φ(t), which are the optimal control gain matrices of COSC, ML-POSC, and ML-DSC, respectively. Functions $\Psi_{ab}\left(t\right)$, $\Pi_{ab}\left(t\right)$, and $\Phi_{ab}\left(t\right)$ are the negative control gains from *b* to *a*. We noted that $\Psi_{ab}\left(t\right)$, $\Pi_{ab}\left(t\right)$, and $\Phi_{ab}\left(t\right)$ are also the negative control gains from *a* to *b* because Ψ(t), Π(t), and Φ(t) are symmetric matrices. While the elements of Ψ(t) related with the memories $z_{1}$ and $z_{2}$ are always 0, those of Π(t) and Φ(t) are not (Figure 4b–f). Thus, the controls of the memories do not appear in COSC, but appear in ML-POSC and ML-DSC. This result indicates that the controls of the memories play an important role in estimation.

We first compares Φ with Ψ in more detail to investigate Φ. $\Phi_{xx}$ and $\Phi_{z^{i}z^{i}}$ are larger than $\Psi_{xx}$ and $\Psi_{z^{i}z^{i}}$ (Figure 4a,d,f), which may decrease $\sum_{xx}$ and $\sum_{z^{i}z^{i}}$. Moreover, $\Phi_{xz^{i}}$ is smaller than $\Psi_{xz^{i}}$ (Figure 4b,c), which may strengthen the positive correlation between *x* and $z^{i}$. Therefore, $\Phi_{xx}$, $\Phi_{z^{i}z^{i}}$, and $\Phi_{xz^{i}}$ may improve estimation, which is consistent with our discussion in Section 5.4. However, $\Phi_{z^{1}z^{2}}$ is larger than $\Psi_{z^{1}z^{2}}$ (Figure 4e), which may weaken the positive correlation between $z^{1}$ and $z^{2}$. It seems to be contrary to our discussion because it may worsen estimation.

In order to understand $\Phi_{z^{1}z^{2}}$, we also compared Φ with Π. Estimation in ML-DSC is more challenging than that in ML-POSC because the controllers cannot completely share their information. Thus, except for $\Phi_{z^{1}z^{2}}$, the absolute values of Φ are larger than those of Π (Figure 4a–d,f) for the same reason as the comparison with Ψ. The problem is only $\Phi_{z^{1}z^{2}}$ (Figure 4e). In ML-POSC, the estimation between $z^{1}$ and $z^{2}$ is not needed because the controllers share their information. As a result, $\Pi_{z^{1}z^{2}}$ is determined only from a control perspective, not an estimation perspective. $\Pi_{z^{1}z^{2}}$ is almost the same with $\Pi_{z^{i}z^{i}}$ (Figure 4d–f), presumably because the cooperative control by the controllers 1 and 2 is more efficient than the independent control. By contrast, in ML-DSC, the estimation between $z^{1}$ and $z^{2}$ is necessary because each controller cannot directly access the other controller. $\Phi_{z^{1}z^{2}}$ is smaller than $\Pi_{z^{1}z^{2}}$ (Figure 4e), which may strengthen the positive correlation between $z^{1}$ and $z^{2}$. Therefore, $\Phi_{z^{1}z^{2}}$ may be determined by a trade-off between control and estimation.

In order to clarify the significance of the decentralized Riccati Equation (Equation 39), we compared the performance of the optimal control function of ML-DSC (Equation 35) with that of the following control functions: (55)$$\begin{array}{cc} \; & u^{i,\Psi }\left(t,z^{i}\right)=-R_{ii}^{-1}B_{i}^{⊤}\left(\Psi \mu +\Psi K_{i}\hat{s}\right), \end{array}$$(56)$$\begin{array}{cc} \; & u^{i,\Pi }\left(t,z^{i}\right)=-R_{ii}^{-1}B_{i}^{⊤}\left(\Psi \mu +\Pi K_{i}\hat{s}\right), \end{array}$$
which replaced Φ with Ψ and Π, respectively. We noted that the first terms are not important because μ(t)=0 is satisfied in this set-up. The result is shown in Figure 5. The variances of the state and the memories are $u^{\Psi }$ > $u^{\Pi }$ > $u^{*}$ (Figure 5a–c). Similarly, the expected cumulative costs are $u^{\Psi }$ > $u^{\Pi }$ > $u^{*}$ (Figure 5d). These orders are consistent with the extent of the optimization of estimation. $u^{\Psi }$ does not take into account estimation at all, and its performance is the worst. $u^{\Pi }$ takes into account only the state estimation, and it performs better than $u^{\Psi }$, but not optimally. $u^{*}$ takes into account the estimation of the other memories and the state, and its performance is optimal.

### 6.2. Two-Dimensional State Case

In this subsection, we conduct a numerical experiment in two-dimensional state case (Figure 6a). In this case, we formulated the target state $x_{t}^{\mathrm{tar}}:=\left(x_{t}^{\mathrm{tar},1},x_{t}^{\mathrm{tar},2}\right)\in R^{2}$, the actual state $x_{t}^{\mathrm{act}}:=\left(x_{t}^{\mathrm{act},1},x_{t}^{\mathrm{act},2}\right)\in R^{2}$, the observation $y_{t}:=\left(y_{t}^{1},y_{t}^{2}\right)\in R^{2}$, and the memory $z_{t}:=\left(z_{t}^{1},z_{t}^{2}\right)\in R^{2}$ as follows (Figure 6b): (57)$$\begin{array}{cc} dx_{t}^{\mathrm{tar},1} & =-2\pi x_{t}^{\mathrm{tar},2}dt, \end{array}$$(58)$$\begin{array}{cc} dx_{t}^{\mathrm{tar},2} & =2\pi x_{t}^{\mathrm{tar},1}dt, \end{array}$$(59)$$\begin{array}{cc} dx_{t}^{\mathrm{act},1} & =\left(-2\pi x_{t}^{\mathrm{act},2}+u_{x,t}^{1}\right)dt+d\omega_{t}^{1}, \end{array}$$(60)$$\begin{array}{cc} dx_{t}^{\mathrm{act},2} & =\left(2\pi x_{t}^{\mathrm{act},1}+u_{x,t}^{2}\right)dt+d\omega_{t}^{2}, \end{array}$$(61)$$\begin{array}{cc} dy_{t}^{1} & =\left(x_{t}^{\mathrm{act},1}-x_{t}^{\mathrm{tar},1}+u_{y,t}^{2}\right)dt+d\nu_{t}^{1}, \end{array}$$(62)$$\begin{array}{cc} dy_{t}^{2} & =\left(x_{t}^{\mathrm{act},2}-x_{t}^{\mathrm{tar},2}+u_{y,t}^{1}\right)dt+d\nu_{t}^{2}, \end{array}$$(63)$$\begin{array}{cc} dz_{t}^{1} & =v_{t}^{1}dt+dy_{t}^{1}, \end{array}$$(64)$$\begin{array}{cc} dz_{t}^{2} & =v_{t}^{2}dt+dy_{t}^{2}, \end{array}$$
where $x_{0}^{\mathrm{tar}}=\left(10,0\right)$, $x_{0}^{\mathrm{act}}∼N\left(x_{0}^{\mathrm{act}}|\left(10,0\right),I\right)$, $y_{0}∼N\left(y_{0}|0,I\right)$, and $z_{0}∼N\left(z_{0}|0,I\right)$. $\omega_{t}^{i}\in R$ and $\nu_{t}^{i}\in R$ are the independent standard Wiener processes. $u_{t}^{i}:=\left(u_{x,t}^{i},u_{y,t}^{i}\right)=u^{i}\left(t,z_{t}^{i}\right)\in R^{2}$ and $v_{t}^{i}:=v^{i}\left(t,z_{t}^{i}\right)\in R$ are the controls of controller *i*. The objective function to be minimized is given as follows:(65)$$\begin{array}{c} J\left[u,v\right]:=E_{u,v}\left[\int_{0}^{10}\sum_{i=1}^{2}\left(10{\left(x_{t}^{\mathrm{act},i}-x_{t}^{\mathrm{tar},i}\right)}^{2}+{\left(u_{x,t}^{i}\right)}^{2}+{\left(u_{y,t}^{i}\right)}^{2}+{\left(v_{t}^{i}\right)}^{2}\right)dt\right]. \end{array}$$The objective of this problem is to minimize the distance of the actual state $x_{t}^{\mathrm{act}}$ from the rotating target state $x_{t}^{\mathrm{tar}}$ by small controls. The solution of Equations (Equation 57) and (Equation 58) is given by $x_{t}^{\mathrm{tar}}=\left(x_{t}^{\mathrm{tar},1},x_{t}^{\mathrm{tar},2}\right)=\left(10\cos\left(2\pi t\right),10\sin\left(2\pi t\right)\right)$. If Equations (Equation 59) and (Equation 60) do not have the standard Wiener processes $d\omega_{t}^{1}$ and $d\omega_{t}^{2}$, respectively, $u_{x,t}^{1}=0$ and $u_{x,t}^{2}=0$ are optimal because $x_{t}^{\mathrm{tar}}=x_{t}^{\mathrm{act}}$ is satisfied. In practice, however, the actual state $x_{t}^{\mathrm{act}}$ does not coincide with the target state $x_{t}^{\mathrm{tar}}$ without the control $u_{x,t}$ due to the state noise $d\omega_{t}$, and needs to be controlled. Controller *i* observes and controls the actual state $x_{t}^{\mathrm{act}}$ only $x_{t}^{\mathrm{act},i}$-axis direction. As a result, the communication between the controllers is more important in this problem.

**Figure 6 entropy-25-00791-f006:**
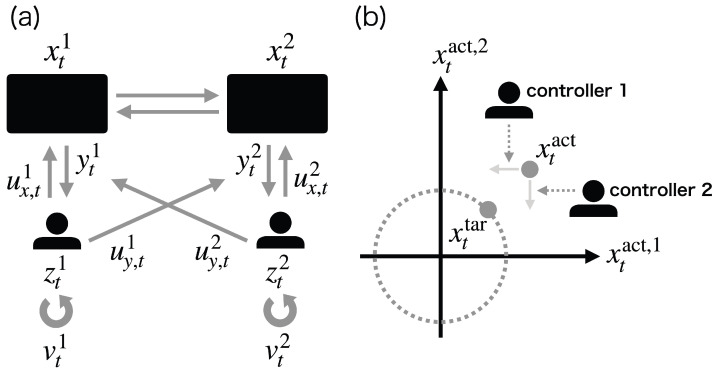
Schematic diagram of the LQG problem in Section 6.2. (**a**) The state of the system $x_{t}=\left(x_{t}^{1},x_{t}^{2}\right)$ is two-dimensional. (**b**) The two controllers control the actual state $x_{t}^{\mathrm{act}}$ to be close to the target state $x_{t}^{\mathrm{tar}}=\left(10\cos\left(2\pi t\right),10\sin\left(2\pi t\right)\right)$. Controller *i* observes and controls the actual state $x_{t}^{\mathrm{act}}$ only $x_{t}^{\mathrm{act},i}$-axis direction.

By defining the state $x_{t}:=x_{t}^{\mathrm{act}}-x_{t}^{\mathrm{tar}}$, Equations (Equation 57)–(Equation 64) are converted as follows: (66)$$\begin{array}{cc} dx_{t}^{1} & =\left(-2\pi x_{t}^{2}+u_{x,t}^{1}\right)dt+d\omega_{t}^{1}, \end{array}$$(67)$$\begin{array}{cc} dx_{t}^{2} & =\left(2\pi x_{t}^{1}+u_{x,t}^{2}\right)dt+d\omega_{t}^{2}, \end{array}$$(68)$$\begin{array}{cc} dy_{t}^{1} & =\left(x_{t}^{1}+u_{y,t}^{2}\right)dt+d\nu_{t}^{1}, \end{array}$$(69)$$\begin{array}{cc} dy_{t}^{2} & =\left(x_{t}^{2}+u_{y,t}^{1}\right)dt+d\nu_{t}^{2}, \end{array}$$(70)$$\begin{array}{cc} dz_{t}^{1} & =v_{t}^{1}dt+dy_{t}^{1}, \end{array}$$(71)$$\begin{array}{cc} dz_{t}^{2} & =v_{t}^{2}dt+dy_{t}^{2}, \end{array}$$
where the initial conditions are given by the standard Gaussian distributions. Furthermore, the objective function (Equation 65) is converted as follows:(72)$$\begin{array}{c} J\left[u,v\right]:=E_{u,v}\left[\int_{0}^{10}\sum_{i=1}^{2}\left(10{\left(x_{t}^{i}\right)}^{2}+{\left(u_{x,t}^{i}\right)}^{2}+{\left(u_{y,t}^{i}\right)}^{2}+{\left(v_{t}^{i}\right)}^{2}\right)dt\right]. \end{array}$$As a result, the problem of controlling the actual state $x_{t}^{\mathrm{act}}$ to be close to the target state $x_{t}^{\mathrm{tar}}$ is equivalent to the problem of controlling the state $x_{t}$ to be close to 0.

In the representation using the extended state $s_{t}:=\left(x_{t}^{1},x_{t}^{2},z_{t}^{1},z_{t}^{2}\right)\in R^{4}$, the extended control ${\tilde{u}}_{t}^{i}:=\left(u_{x,t}^{i},u_{y,t}^{i},v_{t}^{i}\right)={\tilde{u}}^{i}\left(t,z_{t}^{i}\right)\in R^{3}$, and the extended standard Wiener process ${\tilde{\omega }}_{t}:=\left(\omega_{t}^{1},\omega_{t}^{2},\nu_{t}^{1},\nu_{t}^{2}\right)\in R^{4}$ as in Equations (Equation 27) and (Equation 32), the SDEs defined by Equations (Equation 66)–(Equation 71) can be described as follows:(73)$$ds_{t}=\left(\left(\begin{array}{cccc} 0 & -2\pi & 0 & 0\\ 2\pi & 0 & 0 & 0\\ 1 & 0 & 0 & 0\\ 0 & 1 & 0 & 0 \end{array}\right)s_{t}+\left(\begin{array}{ccc} 1 & 0 & 0\\ 0 & 0 & 0\\ 0 & 0 & 1\\ 0 & 1 & 0 \end{array}\right){\tilde{u}}_{t}^{1}+\left(\begin{array}{ccc} 0 & 0 & 0\\ 1 & 0 & 0\\ 0 & 1 & 0\\ 0 & 0 & 1 \end{array}\right){\tilde{u}}_{t}^{2}\right)dt+d{\tilde{\omega }}_{t},$$
which corresponds to Equation (Equation 27). The objective function (Equation 72) can be rewritten as follows:(74)$$J\left[\tilde{u}\right]:=E_{p(s_{0:T};\tilde{u})}\left[\int_{0}^{10}\left(s_{t}^{⊤}\left(\begin{array}{cccc} 10 & 0 & 0 & 0\\ 0 & 10 & 0 & 0\\ 0 & 0 & 0 & 0\\ 0 & 0 & 0 & 0 \end{array}\right)s_{t}+\sum_{i=1}^{2}{\left({\tilde{u}}_{t}^{i}\right)}^{⊤}\left(\begin{array}{ccc} 1 & 0 & 0\\ 0 & 1 & 0\\ 0 & 0 & 1 \end{array}\right){\tilde{u}}_{t}^{i}\right)dt\right],$$
which corresponds to Equation (Equation 32). In addition, it satisfies the block diagonal matrix assumption of R(t) (Equation 31).

Figure 7 shows the trajectories of Ψ(t), Π(t), and Φ(t). Unlike Ψ(t), the elements of Π(t) and Φ(t) related with the memories $z_{1}$ and $z_{2}$ are not always 0 (Figure 7e–j), which indicates that the controls of the memories appear in ML-POSC and ML-DSC. The elements of Φ(t) only largely deviate from those of Π(t) in Figure 7g–j. Figure 7g,h show the negative feedback control gain of the memory $z^{i}$. $\Phi_{z^{i}z^{i}}\left(t\right)$ is larger than $\Pi_{z^{i}z^{i}}\left(t\right)$, which indicates that the memories in ML-DSC are controlled more strongly than those in ML-POSC. Figure 7i shows the control gain between the state $x_{t}^{1}$ and the memory $z_{t}^{2}$. While the controller 1 can control the state $x_{t}^{1}$ based on the memory $z_{t}^{2}$ in ML-POSC, it cannot in ML-DSC, because the controllers do not share their memories in ML-DSC. Furthermore, while controller 1 does not need to send the information of the state $x_{t}^{1}$ to controller 2’s memory $z_{t}^{2}$, this is required in ML-DSC. As a result, $\Phi_{x^{1}z^{2}}\left(t\right)$ differs greatly from $\Pi_{x^{1}z^{2}}\left(t\right)$. A similar discussion is possible for Figure 7j.

In order to clarify the significance of the decentralized Riccati Equation (Equation 39), we compared the performance of the optimal control function $u^{*}$ (Equation 35) with that of the control functions $u^{\Psi }$ (Equation 55) and $u^{\Pi }$ (Equation 56). The result is shown in Figure 8. The actual state $x^{\mathrm{act}}$ faithfully tracks the target state $x^{\mathrm{tar}}$ under the optimal control function $u^{*}$ (Figure 8a–c (green)). Similarly, the memory *z* is stably controlled in the optimal control function $u^{*}$ (Figure 8d,e (green)). As a result, the performance of the optimal control function $u^{*}$ is optimal (Figure 8f (green)).

## 7. Discussion

In this paper, we proposed ML-DSC, which explicitly formulates the finite-dimensional memories of the controllers. In ML-DSC, each controller is designed to compress the infinite-dimensional observation histories appropriately into the finite-dimensional memory, and determines the optimal control based on it. As a result, ML-DSC can handle the difficulty in the conventional DSC that arises from the finiteness of actual memory of controllers. We demonstrated the effectiveness of ML-DSC in the LQG problem. While the conventional DSC needs to restrict the interaction among the controllers to solve the LQG problem, ML-DSC is free from such a restriction. We found that estimation and control are optimized by the decentralized Riccati equation in the LQG problem of ML-DSC. Our numerical experiments showed that the decentralized Riccati equation is superior to the Riccati equation and the partially observable Riccati equation in this problem.

ML-DSC can also address non-LQG problems. In DSC, the non-LQG problem cannot be solved numerically even if the number of the controllers is one, which corresponds to POSC [12,13]. This is because a functional differential equation needs to be solved in the non-LQG problem of POSC, which is generally intractable, even numerically. ML-POSC and ML-DSC are more tractable than the conventional POSC and DSC because only HJB-FP equations need to be solved, which are partial differential equations. The previous work showed that ML-POSC is more effective than the conventional POSC in a non-LQG problem [23]. Therefore, unlike the conventional DSC, ML-DSC may also be effective to non-LQG problems.

In order to solve ML-DSC with a large number of controllers, more efficient numerical algorithms are needed because HJB-FP equations become high-dimensional partial differential equations. In order to solve high-dimensional HJB-FP equations, neural network-based algorithms have been proposed in mean-field stochastic game and control [50,51]. Therefore, by exploiting these neural network-based algorithms, we may efficiently solve ML-DSC with a large number of controllers.

## Figures and Tables

**Figure 1 entropy-25-00791-f001:**
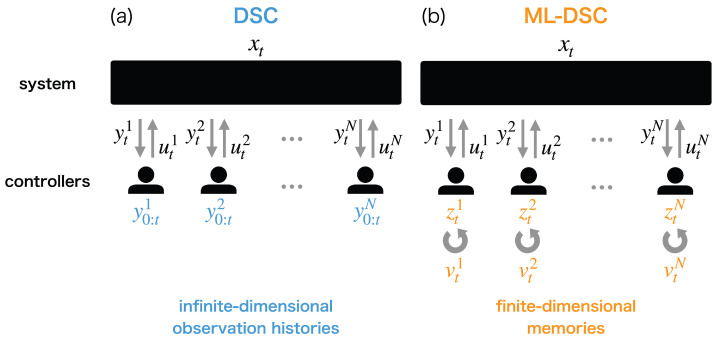
Schematic diagram of (**a**) decentralized stochastic control (DSC) and (**b**) memory-limited DSC (ML-DSC). (**a**) DSC consists of a system and *N* controllers. $x_{t}\in R^{d_{x}}$ is the state of the target system at time t∈[0,T]. $y_{t}^{i}\in R^{d_{y}^{i}}$, $y_{0:t}^{i}:=\left\{y_{\tau }^{i}|\tau \in \left[0,t\right]\right\}$, and $u_{t}^{i}\in R^{d_{u}^{i}}$ are the observation, the observation history, and the control of controller *i*, respectively. The controller *i* cannot accurately observe the state of the system $x_{t}$ and the controls of the other controllers $u_{t}^{j}\;\left(j\neq i\right)$. It can only obtain their noisy observation $y_{t}^{i}$. Then, the controller *i* determines the control $u_{t}^{i}$ based on the noisy observation history $y_{0:t}^{i}$, which ideally requires infinite-dimensional memory to store the observation history $y_{0:t}^{i}$. (**b**) ML-DSC explicitly formulates the finite-dimensional memory $z_{t}^{i}\in R^{d_{z}^{i}}$. Controller *i* compresses the infinite-dimensional observation history $y_{0:t}^{i}$ into the finite-dimensional memory $z_{t}^{i}$ by optimally designing control over the memory $v_{t}^{i}\in R^{d_{v}^{i}}$ as well as control over the state $u_{t}^{i}\in R^{d_{u}^{i}}$.

**Figure 2 entropy-25-00791-f002:**
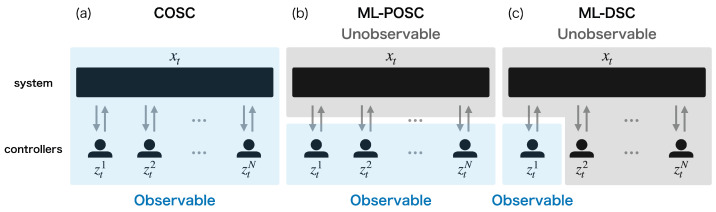
Schematic diagram of (**a**) completely observable stochastic control (COSC), (**b**) memory-limited partially observable stochastic control (ML-POSC), and (**c**) memory-limited decentralized stochastic control (ML-DSC). Blue and gray regions indicate the observables and the unobservables for the controller 1, respectively.

**Figure 4 entropy-25-00791-f004:**
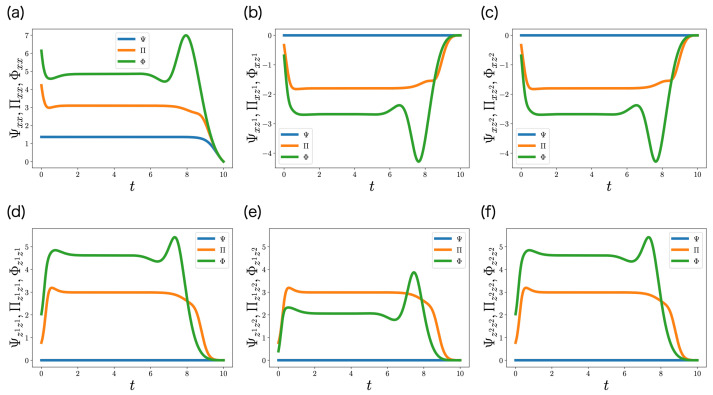
(**a**–**f**) Trajectories of the elements of $\Psi \left(t\right)\in R^{3\times 3}$ (blue), $\Pi \left(t\right)\in R^{3\times 3}$ (orange), and $\Phi \left(t\right)\in R^{3\times 3}$ (green) in Section 6.1. They are the solutions of the Riccati Equation (Equation 38), the partially observable Riccati Equation (Equation 42), and the decentralized Riccati Equation (Equation 39), respectively. Because Ψ(t), Π(t), and Φ(t) are symmetric matrices, $\Psi_{z^{1}x}\left(t\right)$, $\Psi_{z^{2}x}\left(t\right)$, $\Psi_{z^{2}z^{1}}\left(t\right)$, $\Pi_{z^{1}x}\left(t\right)$, $\Pi_{z^{2}x}\left(t\right)$, $\Pi_{z^{2}z^{1}}\left(t\right)$, $\Phi_{z^{1}x}\left(t\right)$, $\Phi_{z^{2}x}\left(t\right)$, and $\Phi_{z^{2}z^{1}}\left(t\right)$ are not visualized.

**Figure 5 entropy-25-00791-f005:**
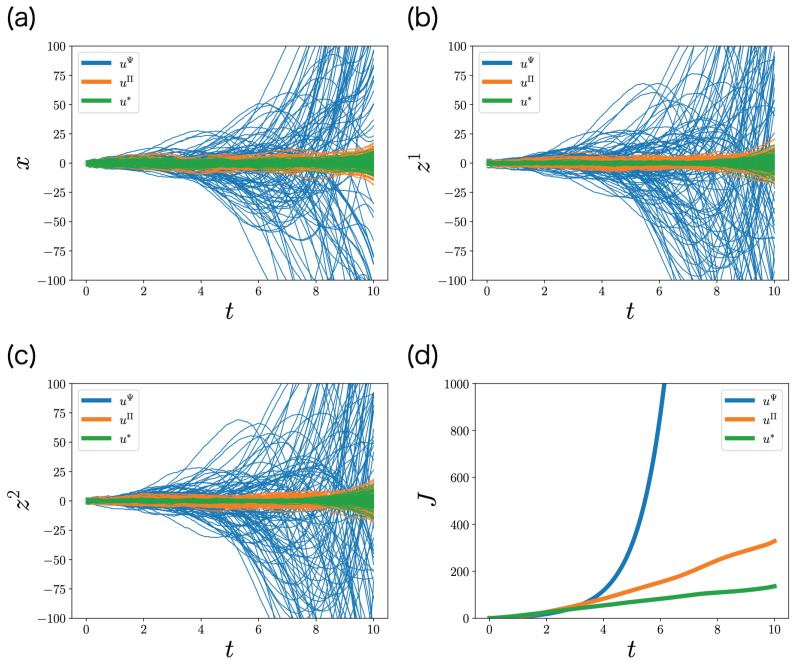
Stochastic simulations in Section 6.1. (**a**–**c**) Stochastic behaviors of (**a**) the state $x_{t}$, (**b**) the controller 1’s memory $z_{t}^{1}$, and (**c**) the controller 2’s memory $z_{t}^{2}$ for 100 samples in Section 6.1. (**d**) The expected cumulative cost computed from the 100 samples. Blue, orange, and green curves are controlled by $u^{\Psi }$ (Equation 55), $u^{\Pi }$ (Equation 56), and $u^{*}$ (Equation 35), respectively.

**Figure 7 entropy-25-00791-f007:**
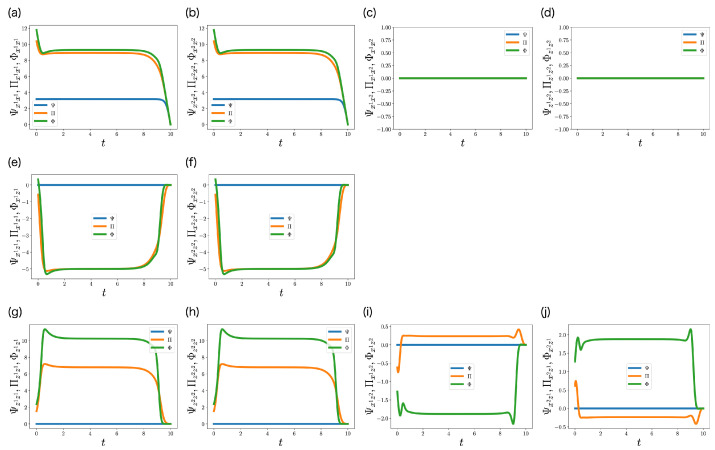
(**a**–**j**) Trajectories of the elements of $\Psi \left(t\right)\in R^{4\times 4}$ (blue), $\Pi \left(t\right)\in R^{4\times 4}$ (orange), and $\Phi \left(t\right)\in R^{4\times 4}$ (green) in Section 6.2. They are the solutions of the Riccati Equation (Equation 38), the partially observable Riccati Equation (Equation 42), and the decentralized Riccati Equation (Equation 39), respectively. Ψ(t), Π(t), and Φ(t) are symmetric matrices, and duplicate elements are not visualized.

**Figure 8 entropy-25-00791-f008:**
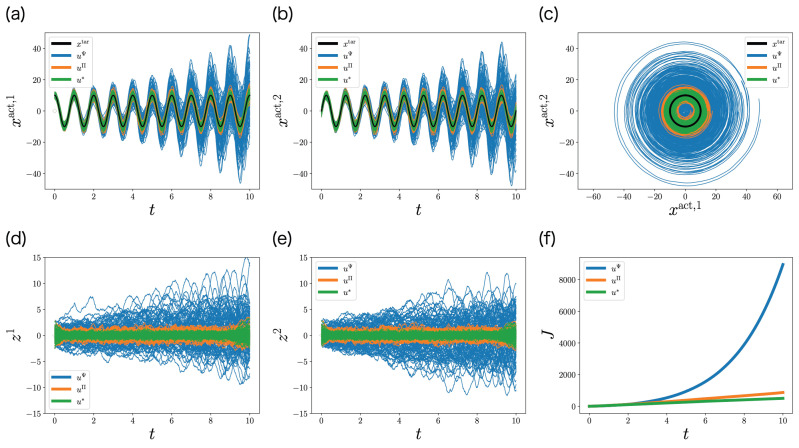
Stochastic simulations in Section 6.2. (**a**–**e**) Stochastic behaviors of the actual state $x_{t}^{\mathrm{act}}=\left(x_{t}^{\mathrm{act},1},x_{t}^{\mathrm{act},2}\right)$ (**a**–**c**) and the memory $z_{t}=\left(z_{t}^{1},z_{t}^{2}\right)$ (**d**,**e**) for 100 samples. (**f**) The expected cumulative cost computed from the 100 samples. Black curves indicate the target state $x_{t}^{\mathrm{tar}}$. Blue, orange, and green curves are controlled by $u^{\Psi }$ (Equation 55), $u^{\Pi }$ (Equation 56), and $u^{*}$ (Equation 35), respectively.

## Data Availability

Not applicable.

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
