# Peer review of "Decentralized Stochastic Control with Finite-Dimensional Memories: A Memory Limitation Approach"

_entropy, 2023, doi:10.3390/e25050791_

Round 1

Reviewer 1 Report

This manuscript presents a detailed analysis of Decentralized stochastic control (DSC) where multiple controllers are unable to accurately observe the target system and the other controllers. The authors propose an alternative theoretical framework based on limited memory (ML-DSC) where each controller is jointly optimized to compress the infinite-dimensional observation history into the prescribed finite- dimensional memory and to determine the control based on it. 

The paper contains novel and interesting results and deserves to be published in Entropy. My only (but strong) advice is to add a figure with a diagram that visually explains all the interconnections between the various SC flavors described in the text.

Reviewer 2 Report

Stochastic control problem formulated in the article consist in decentralized limited memory control of large scale stochastic system.The observations y_t are continuous in time.

1. Formulation of the problem given in section 3 is unclear. What does mean finite dimensional memmory? Real number e.g. \sqrt{2} involve infinite memory. Clarification is needed.

2. Equation (6). Variables v_t^i "is the control of the memory". What kind of control? What is the goal of this control?

3. Theorem 1 gives the conditions of optimality, but it requires solving the HJB equation, which, with the exception of the LQ problem, is practically unfeasible for larger dimensions.   

 4. Theorem 2 is an extension of standard LQG theory

5. The example is too simple to show the characteristic features of the formulated algorithm.

6. The sentence "Each controller can control the memory of the other controller through uiy,t, which can be interpreted as the communication among the controllers." has no connections with equations (47-51).

7.

Round 2

Reviewer 1 Report

I recommend publicatiion of the revised version